# Neutron/Gamma Radial Shielding Design of Main Vessel in a Small Modular Molten Salt Reactor

**Haiyan Yu** [1,2], **Guifeng Zhu** [1,2,*], **Yang Zou** [1,2,*], **Rui Yan** [1,2], **Yafen Liu** [1], **Xuzhong Kang** [1] **and Ye Dai** [1,2]

1   Shanghai Institute of Applied Physics, Chinese Academy of Sciences, Shanghai 201800, China
2   University of Chinese Academy of Sciences, Beijing 100049, China
*   Correspondence: zhuguifeng@sinap.ac.cn (G.Z.); zouyang@sinap.ac.cn (Y.Z.)

**Abstract:** The SM-MSR (small modular molten salt reactor) has a good prospect for development with regards to combining the superiority of the molten salt reactor and modularization technologies, showing the advantages of safety, reliability, low economic cost and flexibility of site selection. However, because its internal structural parts are not easily replaced, and the outer shielding structure is limited, the lifespan of the reactor vessel and its in-reactor shielding design needs to be addressed. In order to find an optimal shielding model with both high fuel efficiency and strong radiation shielding capability, five different design schemes were proposed in this work, which varied in thickness and boron concentration in inner-shielding materials. The neutron/gamma flux and DPA (displacements per atom)/helium production rates were evaluated to obtain an appropriate scheme. Several beneficial results were obtained. Considering the above factors and the actual manufacturing process, 20 cm-thick boron graphite with a 5 wt% Boron-10 concentration combined with a 1 cm-thick Hastelloy barrel was chosen as the in-reactor shielding structure. Outside the reactor, the neutron flux was reduced to $8.33 \times 10^{10}$ cm$^{-2}$ s$^{-1}$, and the gamma flux was decreased to $1.13 \times 10^{11}$ cm$^{-2}$ s$^{-1}$. The vessel/barrel material could maintain a lifespan of more than 10 years, while the burnup depth was 6.25% lower than that of a model without inner-shielding. The conclusions of this research can provide important references for the shielding design and parameter selections of small molten salt reactors in the future.

**Keywords:** small modular molten salt reactor; neutron/gamma flux; radiation damage; burnup calculation; neutron shielding; helium production





## 1. Introduction

Molten salt reactors (MSRs) represent a class of reactors that involve the use of a molten salt either as a coolant in a solid fuel reactor or with nuclear fuel dissolved in molten salt as both fuel and coolant in a liquid-fueled reactor [1,2]. As a candidate of the Generation IV reactors, MSRs are considered to have many merits, such as thorium–uranium breeding ability due to its online refueling, the ability for comprehensive utilization of energy owing to its high temperature output, and even the availability of radioisotopes as auxiliary product from the off-gas system [3].

In the early MSR designs, such as MSBR [2], MSFR [4] and MOSART [5], the original aim is to solve the short supply of nuclear fuel by fuel breeding or to solve long-term disposal of spent fuel by transuranic element transmutation, both by relying on the online reprocessing technology. In recent decades, there has been an increasing interest in MSR design using the once-through fuel cycle mode and modular technologies. This fuel cycle is usually driven by low-enriched uranium instead of U-233, as it has no need for high radiochemical treatment and shows a higher technical maturity and feasibility compared with the breeder or transmutation MSRs. Moreover, the modular technologies adopted in MSRs bring numerous of benefits, such as quality improvement of equipment and safety enhancement, reduction in financial cost and risk and flexibility of site selection [6].

Such SM-MSRs (small modular molten salt reactors) are attractive due to their walk-away safety, reliability, net-zero carbon emissions and low economic cost, and are expected to be commercially deployed with the coming decade. Companies such as Terrestrial Energy [7] and ThorCon [8] have made their R&D plans and are attracting lots of private financing and government funding.

One of the challenges for SM-MSRs is that the materials such as graphite and Ni-based alloys exposed to high neutron flux have short lifespans and those materials immersed in the highly radioactive fuel salt are not easily replaced. Some studies [9,10] on the evaluation and extension analysis of the lifetime of graphite have been carried out. The Ni-based alloy has more than one order of magnitude lower bearable DPA (displacements per atom) than graphite. In addition, there exists helium embrittlement issues of metals under neutron irradiation [11]. Both will lead to an apparent decline in mechanical properties of metal materials. Therefore, the neutron fluence calculation and shielding design of the Ni-based alloy, especially the main vessel used as the pressure and containment boundary, should be given serious attention.

Current reactors generally use heavy elements, including tungsten, lead, stainless steel, etc., to shield gamma, and the lighter materials, including water, concrete, graphite or boron carbide, are used to shield neutrons [12,13]. Most of these shielding designs target the radiation protection outside the reactor building [14,15]; meanwhile, not enough attention has been paid to the shielding analysis of the material inside the reactor. In the PWR (pressurized water reactor), water itself provides good shielding for the main vessel. In the HTGR (high temperature gas-cooled reactor), the permanent reflector block is used to moderate the fast neutron and the side shielding blocks composed of $B_4C/C$, and SUS 316 stainless steel casing is employed as in-reactor shielding [16]. In the MSFR, curved blanket walls are used to reduce the irradiation damages of the outer main vessel [17]. In SM-MSRs, a graphite reflector can also reduce the fast neutron damage of the main vessel, whereas it produces a higher thermal neutron flux, which will produce lots of helium in the main vessel. Usually, one more shielding barrel is required to absorb those thermal neutrons. However, the size requirement of both the graphite reflector and thermal neutron shield barrel in SM-MSRs and their synthetic shielding effects of neutrons and gamma are still unclear.

In this paper, based on the physical model of the SM-MSR designed by SINAP, CAS (Shanghai Institute of Applied Physics, Chinese Academy of Sciences), the thickness of the graphite reflector will be varied to evaluate its effect on fuel utilization and the shielding effect of fast neutrons. Then, the thermal neutron absorbing and gamma shielding ability of the barrel is analyzed by comparing $B_4C$ and Ni-based alloy materials. Finally, the helium production rates in shielding material and the main vessel are calculated in different schemes. It is beneficial to provide an optimal shielding design of SM-MSRs to prolong the lifespan of the main vessel and reduce shielding requirements of the reactor building.

## 2. Model and Methods

### 2.1. Model

The small modular molten salt reactor used in this paper is a thermal spectrum liquid molten salt reactor with a rated thermal power of 150 MWt [18]. $LiF-BeF_2-ThF_4-UF_4$ is used as the fuel and coolant, and the materials of the moderator and reflector are both graphite. The Hastelloy is used as the structure material in the reactor.

The reactor body is composed of the active core, the control rod system, the graphite reflector, upper and lower plenums, a downcomer, inner structural components, which include a fixed bar (hereinafter referred to as the "barrel") and support plate, and a main vessel. Its equivalent height is 370 cm. More details of reactor parameters are listed in Table 1. Figure 1 is the schematic view of the SM-MSR.

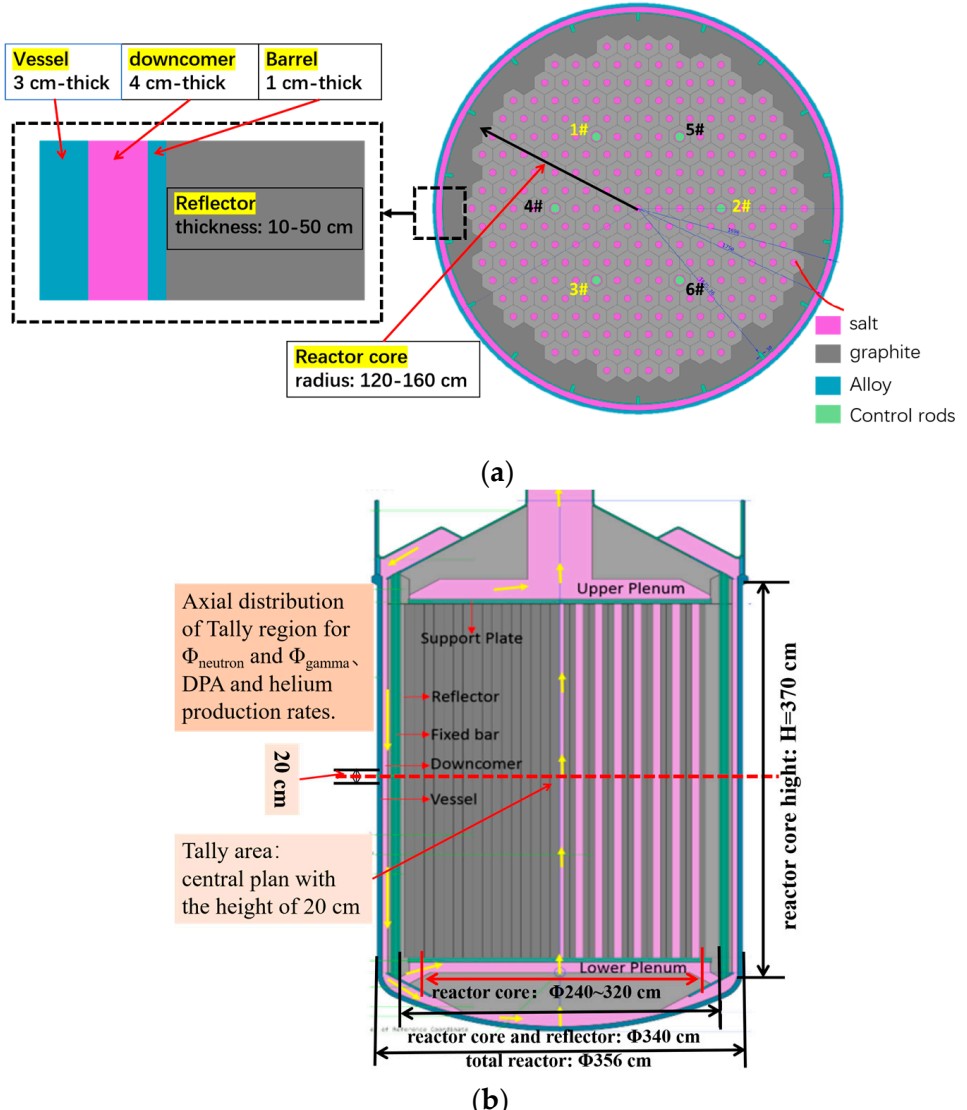

**Figure 1.** The schematic view of the small modular molten salt reactor: (**a**) top view; (**b**) side view.

**Table 1.** Main design parameters of small modular molten salt reactor.

| Parameters | Values |
|---|---|
| Reactor thermal power | 150 MWt |
| Fuel salt | LiF-BeF2-ThF4-UF4 |
| U-235 enrichment | 19.75% |
| Core height of the active zone | 370 cm |
| Active zone radius | 120–160 cm |
| Thickness of radial reflector | 50–10 cm |
| Thickness of reactor barrel | 1 cm |
| Thickness of downcomer | 4 cm |
| Thickness of vessel | 3 cm |
| Outer radius of the stack container | 178 cm |

The modeling interface between the materials is as shown in Figure 1a. The radius of the overall reactor is 178 cm. From the model center to the edge in the radial direction the reactor core, reflector, barrel, downcomer (salt layer) and vessel are constructed, respectively. The total radius of reactor core and side reflector is 170 cm; the radius of the reactor core is from 120 cm to 160 cm, and the thickness of side reflector is from 50 cm to 10 cm. The side

reflector is employed to reflect neutrons to reduce leakage and slow down fast neutrons to reduce fast neutron irradiation. The barrel thickness is 1 cm, and it is the radial support of the reactor to constrain the core components, while forming a downcomer structure with the vessel. The downcomer thickness is 4 cm, and it is the one-piece descending ring cavity of molten salt circulation, which is filled with molten salt to cool the reactor vessel. The vessel thickness is 3 cm; it has the radiation shielding effect and the pressure-bearing effect. The material of the vessel in this paper is Hastelloy.

The shielding effect influenced by the reflector material and barrel material is calculated by tallying the center plane with the height of 20 cm, from the reactor core to the container edge, to obtain neutron/gamma radial flux distribution, DPA and helium production rates results. Additionally, the details of tally areas are shown in Figure 1b.

The goal of the shielding design is to reduce the radiation damage of reactor materials and the neutron/gamma flux outside the reactor as far as possible, which could help to realize the miniaturization of the modular molten salt reactor.

In this paper, the outer diameter of the reactor vessel is maintained as constant, and the structure and materials of the reactor are changed, including reflector thicknesses and barrel materials to evaluate the shielding effect. Here, $B_4C$, Hastelloy and borated materials (borated graphite and borated Hastelloy) are selected as the shielding material. The $B_4C$ or borated material has the advantage of the function of neutron shielding, and the Hastelloy could improve the gamma shielding properties.

Five different design schemes are shown in Table 2. In Scheme A, the shielding effect of the side reflector on neutrons and gamma were studied by changing its thickness from 10 cm to 50 cm; in scheme B and scheme C, the neutrons and gamma shielding effects of the barrel materials (including $B_4C$ and Hastelloy) were compared by flux calculation results; meanwhile, the side reflector was changed from 10 cm to 50 cm.

**Table 2.** Design schemes with different reflector thickness and shielding materials (G stands for graphite, B stands for boron carbide or borated material, H stands for Hastelloy, '-' substitutes separatrix between reflector and barrel).

| Scheme Mark | Reflector | Barrel |
|---|---|---|
| Scheme A: G (thickness)-no barrel | Graphite (G)<br>Thickness change from 10–50 cm | Without barrel (no barrel) |
| Scheme B: G (thickness)-B | Graphite (G)<br>Thickness change from 10–50 cm | $B_4C$ (the boron is natural) (B) |
| Scheme C: G (thickness)-H | Graphite (G)<br>Thickness change from 10–50 cm | Hastelloy (H) |
| Scheme D: BG (concentration, thickness)-H | Graphite+ borated graphite (BG)<br>Total thickness: 35 cm<br>Thickness of borated graphite:5–35 cm<br>($^{10}B$ concentration: 3–30 wt%) | Hastelloy (H) |
| Scheme E: G (35 cm)-BH (concentration) | Graphite (G)<br>Thickness: 35 cm | Borated Hastelloy (BH)<br>$^{10}B$ concentration: 3–30 wt% |

In fact, due to the poor high-temperature mechanical properties of boron carbide and its compatibility with molten salt, there were technical feasibility problems in the boron carbide barrel scheme. For this reason, we further proposed the two schemes of Boron-10 doped in the graphite reflector (scheme D) and Hastelloy (scheme E). In scheme D, the influences of the boron-containing thickness and concentration on the fuel utilization and neutrons/gamma shielding were studied, in this case, the boron-containing thickness was changed from 5 cm to 35 cm (the total thickness of side reflector is 35 cm) and the $^{10}B$ concentration varied from 3 wt% to 30 wt%. In scheme E, we analyzed whether there was a linear relationship between the shielding effects and boron content. In the case of scheme E, the $^{10}B$ concentration in Hastelloy varied from 3 wt% to 30 wt%, and the thickness of the side reflector (without boron added) remains unchanged at 35 cm.

### 2.2. Calculation Methods

The Monte Carlo program MCNP [19] was used for neutron and gamma shielding calculation, and MOBAT program [20,21] was used for burnup calculation under the continuous fuel feeding mode of molten salt reactors.

The burnup depth represents the fuel utilization efficiency, its calculation can be seen in Equation (1) [22]. The fuel in the SM-MSR contains both uranium and thorium, while the unit price of U-235 is much higher than that of thorium. Thus, we only calculated the equivalent burnup depth of uranium to evaluate the fuel economy.

$$\text{Fuel burn} - \text{up (MW·d/KgU)} = \int_0^T P(t)dt/W_U \tag{1}$$

In Equation (1), P(t) is the total power, $T$ is the burnup time, $W_U$ is the total mass of uranium added to the reactor during time $T$, and unit of the burnup depth is MW·d/kgU.

The radiation damage [23,24] includes DPA (displacement per atom) and production rate of transmutation helium gas [25,26], which both impact the life estimation of those structural materials.

The DPA of the compound is weighted by the DPA of the single-element and its component mass percentage, as shown in Equation (2):

$$\text{DPA(total)} = \sum_{i=1}^N w_i \text{DPA}(i) \tag{2}$$

The displacement of atoms rate $R_{dpa}$ is obtained according to Equation (3):

$$R_{dpa}(\text{dpa/s}) = \int_{T_d}^{E_M} \frac{\beta}{2T_d} \sigma_{damage}(E)\Phi(E)dE \tag{3}$$

In Equation (3), $\beta$ is the dislocation efficiency of atoms, which is generally 0.8 [27], and $T_d$ is the energy threshold that causes the dislocation of atoms. The $T_d$ of elements in the reactor material are shown in Table 3. $E_M$ is the maximum neutron energy, $\sigma_{damage}(E)$ is the displacement cross-section for an incident particle at an energy $E$, and $\Phi(E)$ is the neutron flux gained by the calculation results from the Monte Carlo program.

**Table 3.** Atomic displacement threshold energy [28].

| Material | Atomic Displacement Energy in NJOY $T_d$ (eV) |
|:---:|:---:|
| C | 31 |
| Fe | 40 |
| Ni | 39 |
| Mo | 65 |
| Cr | 40 |
| W | 70 |

The helium gas mainly comes from the (n, α) reaction with B-10 and Ni in metal materials. Equation (4) shows the production rate of the transmutation gas helium as follows:

$$\text{GPR (appm He/s)} = \int_0^E \sigma_{(n,\alpha)}(E_i)\Phi(E_i)dE_i \tag{4}$$

$E$ is the maximum neutron energy and $\sigma_{(n,\alpha)}(E_i)$ is the cross-section of the reaction $(n,\alpha)$ at an energy $E_i$.

Helium production rates were calculated by the MOBAT code together with the ORIGEN and MCNP. Materials that evolved with the operation of the reactor were obtained by ORIGEN calculation. Cross-section files and the neutron flux, which changed with

the operation of the reactor, were obtained by MCNP. In order to validate the simulation calculation results, the empirical formula was employed to calculate the helium production rate, and its results are very close to the simulation results.

## 3. Results

### 3.1. The Influence of Barrel Materials on Flux Distribution

Scheme A, B and C were compared in this section to analyze the influence of the barrel materials on the radial flux distribution of the thermal neutron (0–0.05 MeV), fast neutron (0.05–20 MeV) and gamma. The reflector thickness is supposed as 50 cm, and the results are shown in Figure 2.

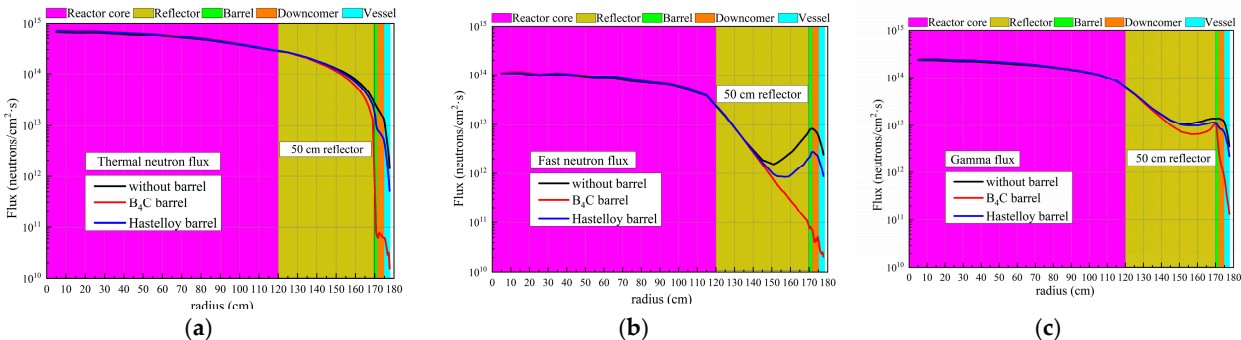

(**a**) (**b**) (**c**)

**Figure 2.** The radial (**a**) thermal neutron, (**b**) fast neutron and (**c**) gamma flux under different barrel materials when the thickness of reflector is 50 cm.

For the thermal neutron flux distribution (Figure 2a), three curves are almost the same in the active core and reflector; however, in the barrel, the downtrends of the curves are different. The flux in the Hastelloy barrel is 2 times lower than that of the scheme without the barrel, and the flux in the $B_4C$ barrel is more than 200 times lower than that of the scheme without the barrel, indicating that boron has a significant ability to absorb thermal neutrons.

For the fast neutron flux distribution (Figure 2b), the performance of the curves in the three schemes are significantly different. In the $B_4C$ barrel scheme, fast neutrons decrease exponentially after graphite moderation, while in schemes without the barrel and with the Hastelloy barrel, the fast neutron flux has a small peak in the downcomer, mainly produced by the fission of the fuel salt in the downcomer. In the $B_4C$ barrel scheme, the thermal neutron in the downcomer is extremely low, so its peak is very small, which shows that $B_4C$ not only has a greater effect on the thermal neutron shielding of the SM-MSR, but also has a significant inhibitory effect on the fast neutron from the downcomer. The fast neutron fluxes in the inner surface of the main vessel with the Hastelloy barrel scheme and with the $B_4C$ barrel scheme are 3 times and 35 times lower than that of the scheme without the barrel, respectively.

For the gamma flux distribution (Figure 2c), there are gamma flux peaks both in the barrel and the molten salt. They mainly come from the boron neutron capture reactions (Equation (5)) and fission reactions of neutron with fuel. At the same time, the heavy metals uranium and thorium in the fuel salt have a good shielding effect on gamma. In scheme A and C, the gamma are produced mainly by the fission nuclear reaction in the fuel salt and the gamma disappear because of the shielding of heavy metals simultaneously. In the $B_4C$ barrel scheme, the number of gamma produced by the fission reaction is fewer due to the fewer thermal neutrons, meanwhile, the gamma are also shielded by heavy metals. Thus, there is a significant drop by an order of magnitude.

$$^{10}B + {}^{1}n \rightarrow {}^{7}Li + {}^{4}He + \gamma \tag{5}$$

In summary, the B$_4$C barrel has a significant shielding effect on fast, thermal neutrons and even gamma, and it is an ideal shielding material for SM-MSRs.

### 3.2. The Influence of Reflector Thickness on Flux Distribution

In this section, we analyze the flux distributions of scheme A, B, and C with different thicknesses of reflector. The thickness of the reflector also has a great effect on the flux distributions of neutron and gamma, as shown in Figures 3–5. The fluxes in the active core increase with the increase in reflector thickness due to the size reduction of the active core and, hence, the rise in power density. In the reflector, the thermal neutrons flux will increase (Figure 3a) and the fast neutron flux and gamma flux will decline as the reflector thickness increases (Figure 3b,c).

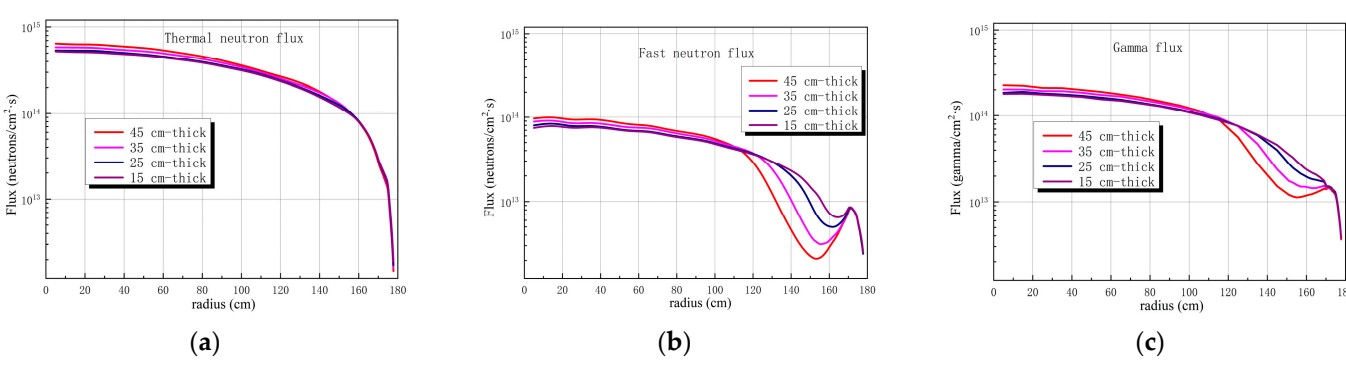

**Figure 3.** Radial fluxes in scheme A (without barrel) with different thicknesses of reflector: (**a**) thermal neutron flux distribution; (**b**) fast neutron flux distribution; (**c**) gamma flux distribution.

In scheme A, without the barrel, as the reflector becomes thicker, more fast neutrons from the active core are moderated into thermal neutrons. On the other hand, the increased thermal neutrons result in more fission reactions in the downcomer, which then produce more fast neutrons. Overall, there is a counteraction between the two mechanisms, leading to little change in the fluxes in the outer side of the vessel (Figure 5a). The error of gamma and neutron flux results in Figure 5 is between 3.51% and 4.25%. The variation tendency of flux (influenced by reflector thickness) is still obvious and it does not affect the judgment of the results.

In scheme B, with the B$_4$C barrel, since boron has a significant ability to absorb thermal neutrons (Figure 4a), and fewer thermal neutrons have fission reaction in the downcomer, fewer fast neutrons and gamma are generated (Figure 4b,c). Thus, the shielding effect increases with the increase in the reflector thickness, as shown in Figure 5b.

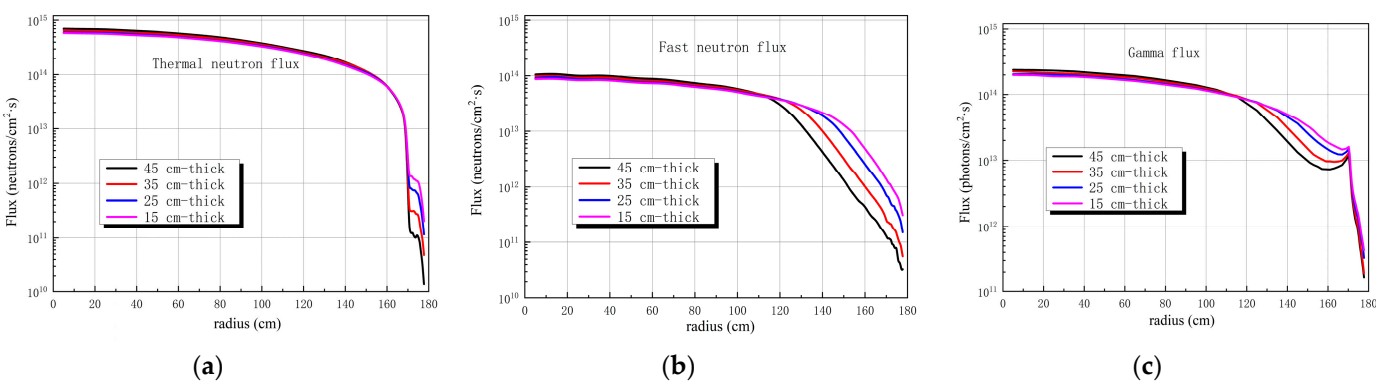

**Figure 4.** Radial fluxes in scheme B (with B$_4$C barrel) with different thicknesses of reflector: (**a**) thermal neutron flux distribution; (**b**) fast neutron flux distribution; (**c**) gamma flux distribution.

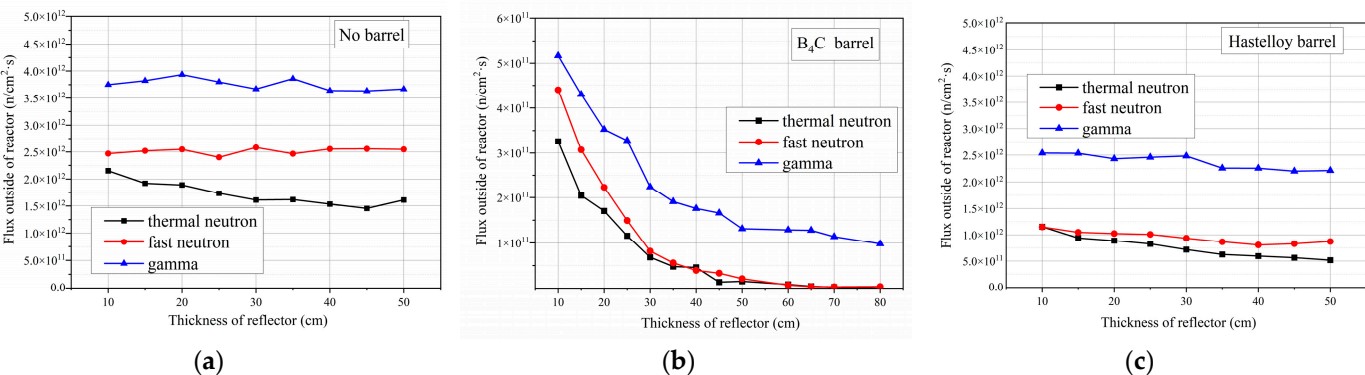

**Figure 5.** The fluxes in the inner surface of the main vessel varied with the reflector thickness: (**a**) scheme A without barrel; (**b**) scheme B with B$_4$C barrel; (**c**) scheme C with Hastelloy barrel.

In scheme C, with the Hastelloy barrel, as shown in Figure 5c, the thermal neutrons are absorbed by the barrel, which leads to the reduction in fission nuclear reaction in the downcomer, and results in the drop of neutron and gamma flux simultaneously. However, its shielding effect is not better than that of scheme B.

Therefore, from the point of view of the shielding effect, the thickness of the reflector should be increased, and boron is a preferable shielding material. From Figure 5b, when the reflector is increased to 35 cm in thickness, the flux of the total neutron is decreased to be $8.5 \times 10^{10}$. However, further increasing the thickness of the reflector does not obviously enhance the shielding effect, but it aggravates the radiation damage of graphite in the active core. In our paper, the reflector thickness chosen was 35 cm in the following research models.

### 3.3. The Influence of Boron Concentration and Distribution on Shielding Effect

Due to the poor high-temperature mechanical properties of B$_4$C and the incompatibility between B$_4$C and molten salt [29], the technical feasibility of using B$_4$C as the barrel directly is doubtful, even cladding B$_4$C with alloy faces the risk of damage. Therefore, we further propose the other two schemes of boron doped in the graphite reflector (scheme D) and Hastelloy barrel (scheme E).

In scheme D, boron graphite is used to replace the outer edge of the reflector in these models. The boron concentration changes from 3% to 30%, and the thickness of boron graphite changes from 5 cm to 35 cm. From the above section analysis, it can be easily understood that the fluxes of neutron and gamma on the outer side of the reactor vessel will decrease as the total boron content increases. In addition, when the total boron content is constant, the fluxes of neutron and gamma on the outer side of the vessel will decrease as the boron is more dispersive (lower boron concentration and larger boron graphite thickness). That is because the space self-shielding effect of boron neutron absorption will weaken under dispersive distribution conditions. When the boron graphite thickness increases from 5 cm to 35 cm with 1000 kg of boron content, the neutron flux will decrease from $7.71 \times 10^{10}$ to $4.43 \times 10^{10}$ cm$^{-2}$ s$^{-1}$ (Figure 6a), and the gamma flux will decrease from $1.51 \times 10^{11}$ to $6.55 \times 10^{10}$ cm$^{-2}$ s$^{-1}$ (Figure 6b).

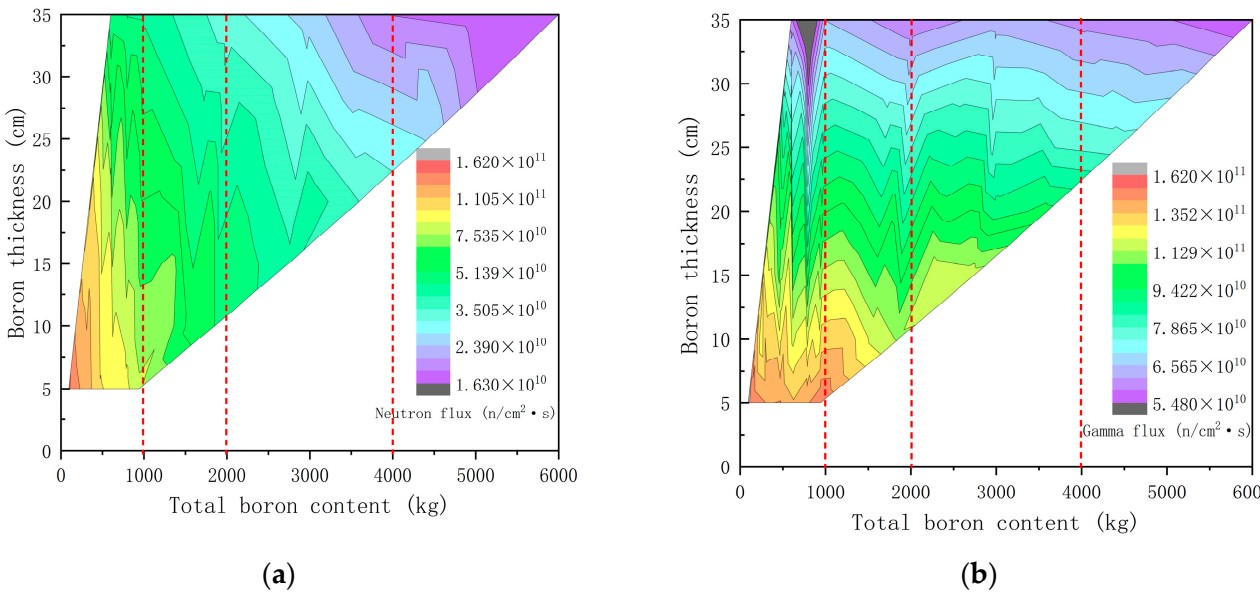

(**a**) (**b**)

**Figure 6.** Total neutron flux (**a**) and gamma flux (**b**) on the outer side of the reactor vessel with different boron contents and boron graphite thickness.

In scheme E, boron is added into a 1 cm-thick Hastelloy barrel, and the boron concentration changes from 3% to 30%. As the boron concentration increases, the neutron shielding effect becomes better. However, there is no significant improvement in the gamma shielding effect, as shown in Figure 7. Here, the error of neutron and gamma flux results in Figure 7 is between 4.08% and 4.45%. To some degree, it could cause the calculation results to not be smoothly presented in picture. However, the trend of the fluxes is clear, and it does not affect the judgment of the shielding effectiveness.

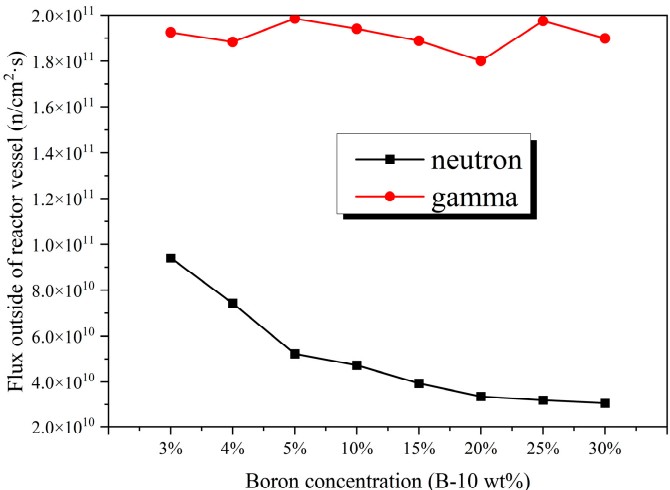

**Figure 7.** Neutron and gamma flux in the inner side of the reactor vessel with different boron contents in the Hastelloy barrel.

In summary, with the same boron content, the shielding effect with the model of boron doped in the graphite reflector is superior to that of boron doped in the Hastelloy barrel, and if the boron distribution is more dispersive, the shielding effect is better.

### 3.4. DPA and He Production Rates in Scheme D/E

The radiation damage of materials in the reactor includes the DPA/He generation rate in the barrel and vessel, and the DPA in the reflector. They are all calculated at the

temperature of 900 K. The DPA and He production rates were calculated in the central cross-section of the barrel and vessel. There is no international standards for molten salt reactors yet; thus, the data from other reactors and the related experiment research [30,31] are cited here. In this paper, the maximum DPA of the vessel and graphite is assumed as 1 dpa and 20 dpa during their lifetime, respectively.

Due to the limitations of current technology concerning processing and manufacturing, boron content in boron graphite is between 3% and 8% [32,33]. Therefore, the effects of models with different thicknesses and B-10 concentrations (3%, 4% and 5%) on radiation damage are studied as follows.

### 3.4.1. DPA in the Barrel and Vessel

In scheme D, DPA in the barrel and vessel are calculated when the thicknesses of boron graphite are 10 cm and 20 cm. It is found that DPA production rates in the barrel and vessel both decrease with the increases in boron graphite thickness and boron concentration. The maximum DPA production rate in the barrel is 0.0028 dpa/year in Figure 8a. For the 1 dpa limit of the alloy, the radiation lifetime of the barrel can reach more than 300 years. In scheme E, the DPA values in the barrel and the vessel is not much different from that in scheme D (Figure 8b).

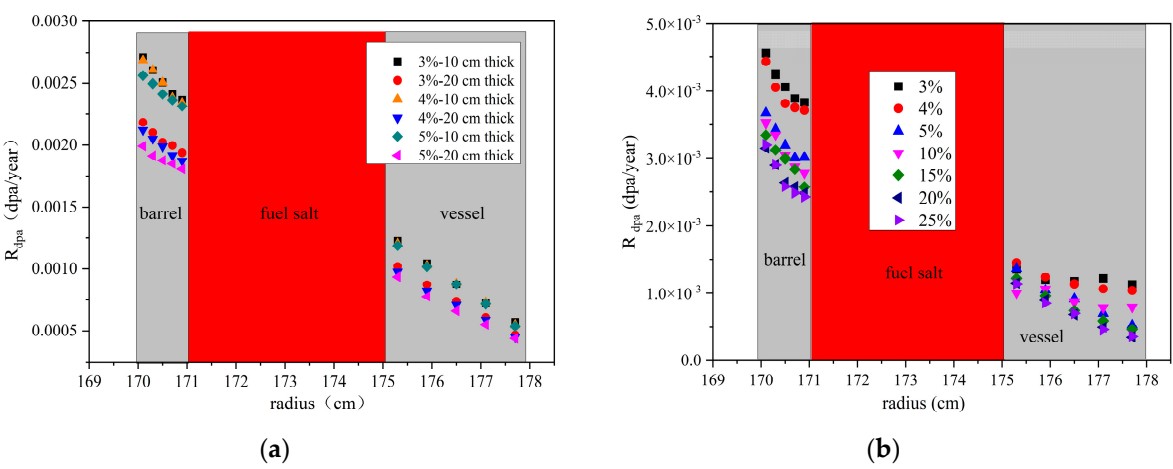

**Figure 8.** DPA in the reactor barrel and vessel of (**a**) scheme D and (**b**) scheme E.

If the boron shielding layer is not taken into account, the maximum DPA production rate in the barrel is 0.03 dpa/year and the radiation lifetime of the barrel is only about 33 years (Figure 9). That implies a little boron doped into either the barrel or the graphite reflector is still required if the reactor lifetime needs to be more than 40 years.

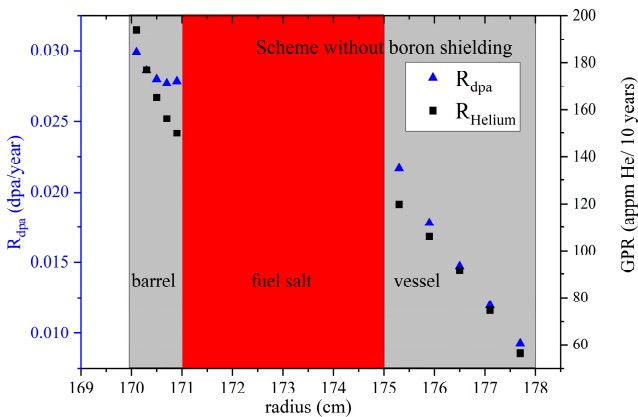

**Figure 9.** DPA and helium production rates in scheme E without boron doped in the barrel.

### 3.4.2. Helium Production in Barrel and Vessel

The helium embrittlement mechanism refers to the embrittlement of helium gas in the alloy, which reduces the ductility of the metal, and it could not be eliminated by high temperature annealing. Therefore, the production of helium in the barrel and vessel composed of alloy materials should be concerned. Natural Ni contains 68% atom $^{58}$Ni and 26% atom $^{60}$Ni, with the fast neutron (n, α) reactions of Equation (6).

$$^{58}\text{Ni} + \text{n} \rightarrow {}^{55}\text{Fe} + {}^{4}\text{He}, \, {}^{60}\text{Ni} + \text{n} \rightarrow {}^{57}\text{Fe} + {}^{4}\text{He} \tag{6}$$

Helium can also be produced by the two step reactions of Equation (7):

$$^{58}\text{Ni} + \text{n} \rightarrow {}^{59}\text{Ni} + \gamma, \, {}^{59}\text{Ni} + \text{n} \rightarrow {}^{56}\text{Fe} + {}^{4}\text{He} \tag{7}$$

However, this reaction requires production of $^{59}$Ni first and results in an incubation time for helium production. At the same time, the reaction of boron with neutrons also produces helium in the alloy. In this article, the boron impurity in the alloy is about 20 ppm, even without doped boron.

In scheme E, without doped boron in the barrel, the total helium production at the ten year stage is less than 200 appm in the barrel, as shown in Figure 9. The helium production in the barrel is mainly contributed to by Ni-59; while in the reactor vessel with lower neutron flux, it mostly comes from the boron reaction since the mass of Ni-59 has not yet built.

When boron is doped in the graphite, as given in scheme D, it is found that helium production in the barrel and vessel both decrease with the increase in boron graphite thickness and boron concentration (Figure 10a), almost is an order of magnitude lower than that without doped boron.

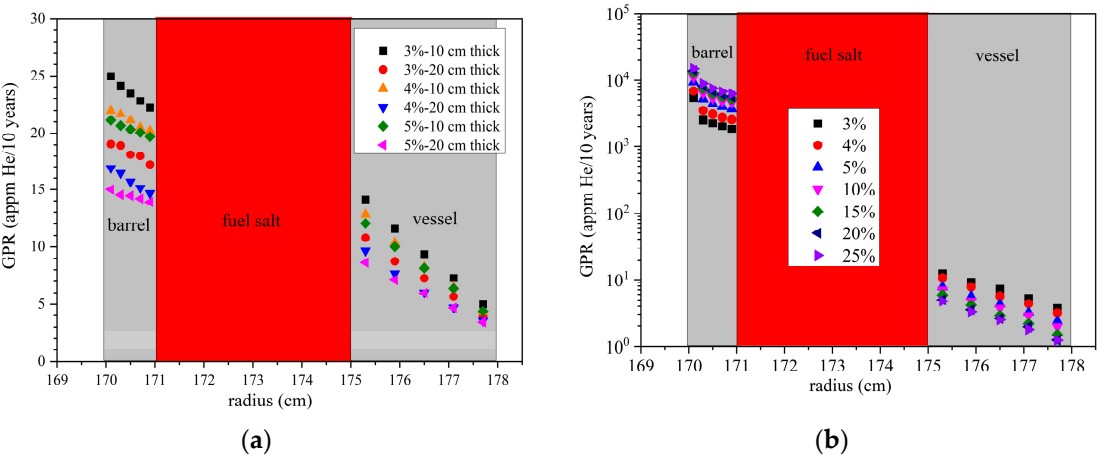

**Figure 10.** Helium production rates in (**a**) scheme D and (**b**) scheme E with different boron contents.

When boron is added into the alloy barrel, as given in scheme E, the helium production in the barrel is from approximately 2000–10,000 appm, changing with the boron content (Figure 10b), which is mainly contributed to by the boron neutron capture reaction. It is worth noting that the helium production is not linear with the doped boron content since the boron itself has a self-shielding effect. There is no international standards for helium production rate limitations for molten salt reactors yet; thus, the data from other related experimental research are referred to. According to the previous research [34], the helium in the alloy should not exceed 2000 appm, otherwise it can seriously embrittle the barrel. Therefore, scheme E is not suitable for the shielding design for the material damage of the barrel could not be ignored.

### 3.4.3. DPA in Graphite

In scheme D, boron concentration remains at 5%; if the thickness of the boron graphite is larger, the DPA production rate of the graphite in the active core will be larger. The reason is that when the boron is closer to the active core, the boron will have a more obvious absorption of thermal neutrons, which should be reflected into the active core, hence, making the fission more concentrated upon the center of the active core. When the thickness of boron graphite is 35 cm, the maximum graphite DPA is 2.2 dpa/year. If boron-10 is not doped in the graphite reflector, the maximum graphite DPA is approximately 1.2 dpa/year, and the change in $R_{dpa}$ values from the center to the margin in the graphite area will be smoother (Figure 11a).

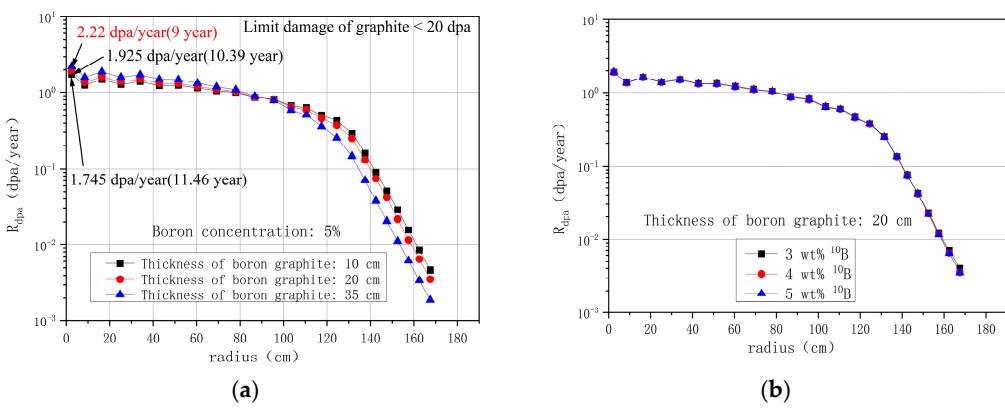

**(a)**            **(b)**

**Figure 11.** DPA in the graphite of (**a**) scheme D and (**b**) scheme E.

In order for active graphite to maintain a lifespan of more than 10 years, the thickness of the boron graphite should be less than 20 cm. In addition, $R_{dpa}$ values of graphite in the active core were kept almost unchanged with the increase in boron concentration in the barrel, as shown in Figure 11b. Thus, from the view of reducing the maximum $R_{dpa}$ value of graphite, the scheme with a boron graphite thickness of less than 20 cm and a high boron concentration is preferable.

### 3.5. Analysis of Fuel Utilization under Different Shielding Schemes

The burnup program MOBAT was used to find the fuel utilization change under different schemes.

In scheme A, it is found that the increased thickness of the reflector slightly lowers the fuel burnup, as shown in Figure 12a. The fuel burnup is approximately 200 MW·d/kgU at the 10 full power years (fpy).

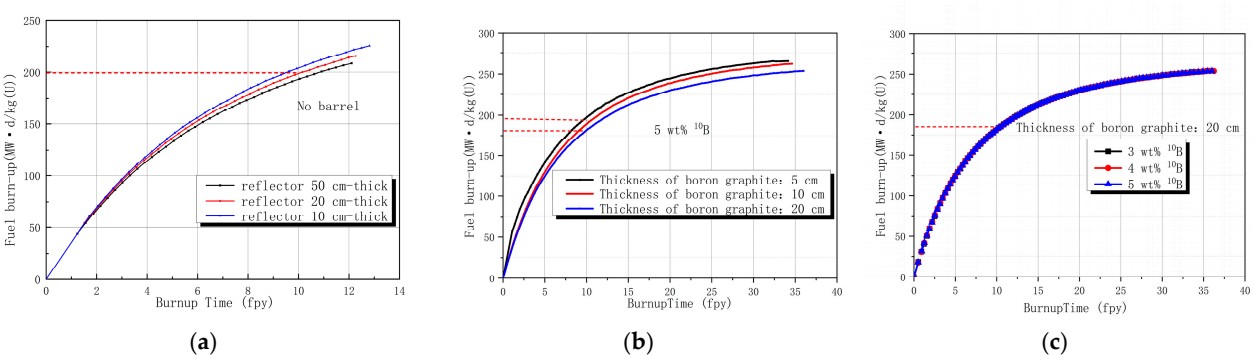

**(a)**            **(b)**            **(c)**

**Figure 12.** Burn-up depth in (**a**) scheme A, (**b**) scheme D with changed thickness of boron graphite and (**c**) scheme D with changed boron concentration.

In scheme D, as shown in Figure 12b,c, with the increased thickness of the boron graphite, the burnup depth declines, while the boron concentration has little effect on the burnup. Under the model with 20 cm-thick boron graphite (5 wt%$^{10}$B), the fuel burnup depth is 187.5 MW·d/kgU, which is 6.25% lower than that of the model without boron graphite.

## 4. Conclusions

The shielding design is an essential issue of reactor safety. In this research, the radiation shielding effects and radiation damage of key materials of a reactor were evaluated under the different proposed design schemes.

According to our research, we could draw the following conclusions: (1) if there is no inner-shielding, the barrel and vessel can maintain a service life of 10 years, but not meet 40 years; (2) boron plays an important role in reducing fast neutrons, thermal neutrons and gamma, so it is necessary to set up a boron absorbing layer; (3) boron should not be doped in the alloy, which would cause more serious helium embrittlement problems, therefore, boron doped into the graphite reflector is recommended; (4) boron doped into the graphite reflector has a certain deterioration effect on the DPA of graphite in the active core and burnup depth. It is suggested to reduce the thickness of the boron reflection layer on the premise of satisfying the shielding effect.

In this paper, we recommend scheme D, comprising 20 cm-thick boron graphite with 5% B-10 concentration, which replaces the 20 cm-thick reflector at the outer edge of the graphite reflector, and the total thickness of the graphite and borated graphite is 35 cm. The final radial structure of the reactor consists of a reactor core with a radius of 135 cm, a 15 cm-thick graphite reflector, a 20 cm-thick boron graphite shielding layer, a 1 cm-thick Hastelloy barrel, a 4 cm-thick downcomer and a 3 cm-thick vessel. In the situation of the above shielding design, the neutron flux outside the reactor is $8.33 \times 10^{10}$ cm$^{-2}$ s$^{-1}$, which is approximately 50 times lower compared with a neutron flux of $3.86 \times 10^{12}$ cm$^{-2}$ s$^{-1}$ if there is no shielding design. It provides a reference for the shielding design and parameter selections of a small modular molten salt reactor, while the details of the parameters may be changed as the shielding requirements change.

In the future, there is still further work to be confirmed and verified, such as the mechanical and thermal stability properties of the proposed shielding material doped with B-10 and their compatibility with fuel salt. Additionally, burnable poison without helium production, such as Gd, can be tried as the neutron shielding material directly doped into the Hastelloy, which may have the advantage of less harm to the neutron value of the reactor core and less irradiation damage to the Hastelloy itself.

**Author Contributions:** Conceptualization, H.Y. and G.Z.; methodology, H.Y. and Y.L.; software, Y.D. and G.Z.; validation, Y.L. and X.K.; formal analysis, H.Y.; investigation, H.Y.; resources, G.Z.; data curation, H.Y.; writing—original draft preparation, H.Y.; writing—review and editing, G.Z., R.Y. and X.K.; visualization, H.Y.; supervision, Y.Z.; project administration, Y.Z.; funding acquisition, G.Z. All authors have read and agreed to the published version of the manuscript.

**Funding:** This research was funded by the National Natural Science Foundation of China (Grant/Award Number: 12005290); Shanghai Sailing Program (Grant/Award Number: 19YF1457900); Youth Innovation Promotion Association of the Chinese Academy of Sciences (Grant/Award Number: 2020261); the Chinese TMSR Strategic Pioneer Science and Technology Project (Grant/Award Number: XDA02010000).

**Data Availability Statement:** The data presented in this study are openly available in doi:10.13140/RG.2.2.23586.30405.

**Conflicts of Interest:** The authors declare no conflict of interest.

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
