# Peer review of "Neutron/Gamma Radial Shielding Design of Main Vessel in a Small Modular Molten Salt Reactor"

_jne, doi:10.3390/jne4010017_

Round 1

Reviewer 1 Report

This paper presents the results of numerical investigation into vessel in small modular molten salt reactor. The topic of this paper is interesting but its content should be improved. The detailed comments are as follows:

l  For model validation, comparison of the results obtained from the present model and the data reported in literature or measured in field test should be given clearly. Suggest that the comparative results be presented in table and/or figure forms.

l  The critical variables and their limit values for shielding design stipulated in international standards should be added, and their application to this work should be addressed.

l  Input variables and their numerical values used in the simulation should be presented.

l  Modeling the interface between the materials (e.g. barrel, salt, vessel) of the reactor should be discussed.

l  Explains the reason why the shielding materials in table 2 are selected in this study.

l  Improve the resolution and enlargement of figures for better readability.

Author Response

Reviwer1#:

This paper presents the results of numerical investigation into vessel in small modular molten salt reactor. The topic of this paper is interesting but its content should be improved. The detailed comments are as follows:

  • For model validation, comparison of the results obtained from the present model and the data reported in literature or measured in field test should be given clearly. Suggest that the comparative results be presented in table and/or figure forms.

Response1:We feel great thanks for your professional review work on our article. The small modular molten salt reactor in this manuscript is different from the other present models, including the reactor type and the reactor size. There may be not comparable for our model and other models. Thus, sorry for that we didn’t present the comparison. While the results shown in this paper had been well checked. The neutron and gamma fluxes were calculated by the common Monte Carlo transportation code, which has high calculation accuracy even with any complicated geometry model. The DPAs were calculated by equation (3), which is widely used in other references, and we had checked that we have the similar results in the same case. The helium production was calculated by molten salt burnup code MOBAT. The reaction chains were checked, and the main contribution comes from equation (7), which is the similar to that in other reactors. We are confident about the results, and some descriptions are added into the paper in the “2.2. Calculation Methods”.

 It is shown as the following:

Helium production rates is calculated by the MOBAT code, coupling with the ORIGEN and MCNP. Materials that evolves with the operation of the reactor were obtained by ORIGEN calculation; cross-section files and the neutron flux, changed with the operation of reactor, are obtained by MCNP. In order to validate the simulation calculation results, the empirical formula is employed to calculate the helium production rate, and its results are very close to the simulation results.

  • The critical variables and their limit values for shielding design stipulated in international standards should be added, and their application to this work should be addressed.

Response2: Thanks for your great suggestion of our manuscript. Molten salt reactor is a kind of novel nuclear reactor with less operation experience. The irradiation data both of graphite and nicker base alloy are sorely lacking. The requirements of material bearing the different functions will be different. Thus, there is no such an international standards yet. While, we had tried to find the data from other reactors, and had discussed the limit requirements in the paper. We supposed that the limit of graphite DPA, the vessel DPA and the helium production vessel are 20 dpa, 1 dpa, and 2000 ppm. Based on this assumption, we obtained the lifespan of materials and gave the recommended shielding scheme.

Which is modified in revised paper in part of “3.4. DPA and He production rates in Scheme D/E”, specific content is shown as following:

“3.4. DPA and He production rates in Scheme D/E

The radiation damage of materials in the reactor includes the DPA/He generation rate in the barrel and vessel, and the DPA in the reflector. They are all calculated at the temperature of 900 K. The DPA and He production rates were calculated in the central cross section of the barrel and vessel. For there is no an international standards for molten salt reactor yet, thus, the data from other reactors and related experiment research [30, 31] are cited here. In this paper, it assumed that the maximum DPA of Hastelloy alloy as the vessel and graphite are 1 dpa and 20 dpa during the lifetime , respectively.”

3.4.2. Helium production in barrel and vessel

When boron is added into the alloy barrel, as given in scheme E, the helium production in the barrel is about 2000-10000 appm changed with the boron content (Figure 10b), which is mainly contributed by boron neutron capture reaction. It’s worth noting that the helium production is not linear with the doped boron content, since the boron itself has self-shielding effect. For there is no an international standards of helium production rates limitation for molten salt reactor yet, thus, the data from other related experiment research are referred to. According to the previous research [34], the helium in the alloy should not exceed 2000 appm, otherwise it can embrittle the barrel seriously. Therefore, scheme E is not suitable for the shielding design for its material damage of barrel could not be ignored.

  • Input variables and their numerical values used in the simulation should be presented.

Response3:Thank you for your kind reminder. We have modified the Table 2, and added the variables and their numerical values used in the simulation.

We list it as following

Table 2. Design schemes with different reflector thickness and shielding materials (G stands for graphite, B stands for boron carbide or borated material, H stands for Hastelloy,-substitutes separatrix between reflector and barrel)

Five different design schemes are shown in Table 2. In Scheme A, shielding effect of the side reflector on neutrons and gamma were studied by changing its thickness from 10 cm to 50 cm; In scheme B and scheme C, the neutrons and gamma shielding effects of barrel materials (including B4C and Hastelloy) were compared by flux calculation results, and meanwile, the side reflector is changed from 10 cm to 50 cm.

In fact, due to the poor high-temperature mechanical properties of boron carbide and its compatibility with molten salt, there are technical feasibility problems in the boron carbide barrel scheme. For this reason, we further proposed the two schemes of Boron-10 doped in the graphite reflector (scheme D) and Hastelloy (scheme E). In the scheme D, the influences of the boron-containing thickness and concentration on the fuel utilization and neutrons/gamma shielding were studied, in this case, the boron-containing thickness is changed from 5 cm to 35 cm (the total thickness of side reflector is 35 cm ) and the 10B concentration varies from 3 wt% to 30 wt%; in the scheme E, we analyzed that whether there is a linear relationship between the shielding effects and boron content. In the case of scheme E, the 10B concentration in hastelloy varies from 3 wt% to 30 wt%, and the thickness of side reflector(without boron added) remains 35 cm unchanged.

4Modeling the interface between the materials (e.g. barrel, salt, vessel) of the reactor should be discussed.

Response4:Thank you for your careful review. We have added the description of modeling the interface between materials in “2. Model and Methods” part, which is list as following:

“The modeling interface between the materials is as shown in Figure 1a. The radius of overall reactor is 178 cm. From the model centre to the edge in the raidal direction: reactor core, reflector, barrel, downcomer (salt layer) and vessel are constructed, respectively. The total radius of reactor core and side reflector is 170 cm, for the radius reactor core is from 120 cm to 160 cm, and the thickness of side reflector is 50 cm to 10 cm, respectively. The side reflector is employed to reflect neutrons to reduce leakage, and slow down fast neutrons to reduce fast neutron irradiation. The barrel thickness is 1cm and it is the radial support of reactor to constrain the core components, and meanwhile, forming a downcomer structure with the vessel. The downcomer thickness is 4 cm, and it is the one-piece descending ring cavity of molten salt circulation, which is filled with molten salt to cool the reactor vessel. The vessel thickness is 3 cm, it has the radiation shielding effect and the pressure-bearing effect, and the material of vessel in this paper is hastelloy.

The shielding effect influenced by reflector material and barrel material is calculated, by tallying the centre plane with the height of 20 cm, from the reactor core to the container edge, to obtain neutron/gamma radial flux distribution, DPA, and helium production rates results. Additionally, the details of tally areas are shown in Figure 1b.

The goal of the shielding design is to reduce the radiation damage of reactor materials and the neutron/gamma flux outside the reactor as far as possible, which could help to realize the miniaturization of modular molten salt reactor.”

5Explains the reason why the shielding materials in table 2 are selected in this study.

Response5: We think this is an excellent suggestionAs suggested by the reviewer, we have explained in the revised manuscipt in the part of “2. Model and Methods”.

The following is the addition explanation:

“In this paper, the outer diameter of the stack container is maintained constant, and the structure and materials of the reactor are changed, including reflector thicknesses and barrel materials to evaluate the shielding effect. Here, B4C, Hastelly and borated materials (borated graphite and borated hastelloy) are selected as the shielding material. For the B4C or borated material has the advantage function of thermal neutron shielding, and the hastelloy could improve the gamma shielding properties and themechanical properties of the reactor container structure. ”

Reviewer 2 Report

The paper deals with the investigation of the effectiveness of graphite reflector with different thickness on fuel utilization and the shielding effect of fast neutron. The topic of the manuscript is worthy of investigation and well fits with the scope of the journal. The presentation of the background is good and the results are well discussed.

This reviewer thinks the paper can be accepted for publication in JNE in its current form.

Author Response

Reviwer2#

The paper deals with the investigation of the effectiveness of graphite reflector with different thickness on fuel utilization and the shielding effect of fast neutron. The topic of the manuscript is worthy of investigation and well fits with the scope of the journal. The presentation of the background is good and the results are well discussed.

This reviewer thinks the paper can be accepted for publication in JNE in its current form.

Response:Thank you for your very encouraging comments on the merits. I would like to express our sincere appreciations of your letter.

Reviewer 3 Report

The development of new designs of molten salt reactors including their miniaturization is necessary for increasing the efficiency of the power plants. In the paper "Neutron/gamma radial shielding design of the main vessel in a small modular molten salt reactor" the new shielding design was developed for small molten salt reactors. The authors have made well theoretical work and discussed the results. However, in my opinion, some points of the paper should be improved accordingly following comments:

1.                  The authors should discuss the safety of small reactors. The decrease in size leads to the necessity of the increasing amount of reactors to produce the same amount of energy. It increases the probability of accidents.

2.                  The authors have not discussed the mechanical properties and thermal stability of the developed shielding. The properties cannot be fully determined by Hastelloy due to the salt interlayer.

3.                  A minor correction is also recommended:

- Title of Part 4 should be changed from Discussion to Conclusions.

Author Response

Reviwer3#

The development of new designs of molten salt reactors including their miniaturization is necessary for increasing the efficiency of the power plants. In the paper "Neutron/gamma radial shielding design of the main vessel in a small modular molten salt reactor" the new shielding design was developed for small molten salt reactors. The authors have made well theoretical work and discussed the results. However, in my opinion, some points of the paper should be improved accordingly following comments:

  • The authors should discuss the safety of small reactors. The decrease in size leads to the necessity of the increasing amount of reactors to produce the same amount of energy. It increases the probability of accidents.

Response1: Thank you for your constructive and insightful suggestions. At present, we are still in the preliminary design stage, in the next stage, we will do further research on safety performance based on our optimized shielding design structure proposed in our paper, because the increase in the number of small reactor will to some extent increase safety risks. It's worth mentioning that the safety of a single small reactor is generally considered to be improved, when compared to a large reactor. Mainly because of the improvement of equipment manufacturing quality in the factory processing, for small reactor has made a lot of simplification in the design. For instance, reducing power density and simplifying system complexity, which improve the ratio of passive safety performance, and could reach the goal of no large-scale radioactive release.

  • The authors have not discussed the mechanical properties and thermal stability of the developed shielding. The properties cannot be fully determined by Hastelloy due to the salt interlayer.

Response2:We are grateful for the valuable and professional suggestions.

In this paper, from the perspective of reactor core design, advanted shielding design is proposed to reduce the irradiation intensity of the reactor material. The mechanical properties and the thermal stability of the developed shielding have been listed for the further research. Furthermore, we will cooperate with materials scientists and engineering mechanics experts to evaluate mechanical properties and the thermal stability based on our provided irradiation flux data, experimental irradiation, and corrosion data.

In this paper, the properties of hastelloy doped with B-10 is not mainly concerned, but it indeed needs to be further investigated. The compatibility of such hastelloy with molten salts needs to be verified carefully. At present, in our study, the hastelloy employed in paper is refer to other reactor, and its manufacturing process and material application have been took into consideration.

We have added a preliminary prospect in the end of our paper, which is as following:

“In the future, there is still further work to be confirmed and verified, such as the me-chanical and the thermal stability properties of the proposed shielding material doped with B-10 and their compatibility with fuel salt. Also, burnable poison without helium production, like Gd, can be tried as the neutron shielding material directly doped into the hastelloy, which may have the adavantge of less harm to the neutron value of reactor core and less irradation damage to the hastelloy itself. ”

Thank you for your professional suggestion again.

3A minor correction is also recommended:

- Title of Part 4 should be changed from Discussion to Conclusions.

Response3:Thanks you for your suggestion. We have corrected the title of part 4 from “Discussion” to “Conclusion”.

Reviewer 4 Report

Paper presents simulations studies on the SM-MSR reactor shieldings use for neutron and gamma flux suppression. As MSR reactors are today seriously taken into account as next generation reactors, this topic is worth to be published. moreover, it should be find a lot of interest among journal readers.
This paper is one of the steps in the way of designing optimal shielding for SM-MSR reactors.

Before publicatin I would like to ask some questions, and suggest improvements, as:
1) Table 2, schemes are explained here - please improve readibility of that Table,  in Reflector collumn only schemes B and D have variables thicknes of the graphite? - then why in Scheme mark there it G(thickness). Please re-arrange it, event with putting more text inside the table.

2)please provide reference for generally constant beta value in equation (3)  - line 132,

3) Figure 2  why on the Figures reflector starts from 0 up to 170 cm in radius, while in the text it is written that it is 50 cm thick? Please indicate properly reflector and core on all of the Figures. As it is now with marked reflector from 0 up to 170 cm, sudden change of fast neutrons fluxes in  at ~150 cm looks strangly.

4) Please write what it estimated uncertainty of the calculations, this can lead to the not smooth behaviour of the calculated values in the figure 5 and 7.

Author Response

Reviwer4#

Paper presents simulations studies on the SM-MSR reactor shieldings use for neutron and gamma flux suppression. As MSR reactors are today seriously taken into account as next generation reactors, this topic is worth to be published. moreover, it should be find a lot of interest among journal readers.
This paper is one of the steps in the way of designing optimal shielding for SM-MSR reactors.

Before publicatin I would like to ask some questions, and suggest improvements, as:
1) Table 2, schemes are explained here - please improve readibility of that Table,  in Reflector collumn only schemes B and D have variables thicknes of the graphite? - then why in Scheme mark there it G(thickness). Please re-arrange it, event with putting more text inside the table.

Response1: Thank you for your good suggestion, we have modified the Table 2 in our revised manuscript, and added some description to improve readibility of table 2, we have list it as following.

Table 2. Design schemes with different reflector thickness and shielding materials (G stands for graphite, B stands for boron carbide or borated material, H stands for Hastelloy,-substitutes separatrix between reflector and barrel)

Scheme mark

Reflector

Barrel

Scheme A: G(thickness)-no barrel

Graphite(G)

Thickness change from 10-50 cm

Without barrel(no barrel)

Scheme B: G(thickness)-B

Graphite(G)

Thickness change from 10-50 cm

B4C (the boron is natural) (B)

Scheme C: G(thickness)-H

Graphite(G)

Thickness change from 10-50 cm

Hastelloy(H)

Scheme D: BG(concentration, thickness)-H

Graphite+ borated graphite(BG)

Total thickness: 35 cm

Thickness of borated graphite:5cm-35cm

(10B concentration: 3 wt%-30 wt%)

Hastelloy(H)

Scheme E: G(35cm)-BH(concentration)

Graphite(G)

Thickness: 35 cm

Borated Hastelloy(BH)

10B concentration:3wt%-30wt%

Five different design schemes are shown in Table 2. In Scheme A, shielding effect of the side reflector on neutrons and gamma were studied by changing its thickness from 10 cm to 50 cm; In scheme B and scheme C, the neutrons and gamma shielding effects of barrel materials (including B4C and Hastelloy) were compared by flux calculation results, and meanwile, the side reflector is changed from 10 cm to 50 cm.

In fact, due to the poor high-temperature mechanical properties of boron carbide and its compatibility with molten salt, there are technical feasibility problems in the boron carbide barrel scheme. For this reason, we further proposed the two schemes of Boron-10 doped in the graphite reflector (scheme D) and Hastelloy (scheme E). In the scheme D, the influences of the boron-containing thickness and concentration on the fuel utilization and neutrons/gamma shielding were studied, in this case, the boron-containing thickness is changed from 5 cm to 35 cm (the total thickness of side reflector is 35 cm ) and the 10B concentration varies from 3 wt% to 30 wt%; in the scheme E, we analyzed that whether there is a linear relationship between the shielding effects and boron content. In the case of scheme E, the 10B concentration in hastelloy varies from 3 wt% to 30 wt%, and the thickness of side reflector(without boron added) remains 35 cm unchanged.”

2) Please provide reference for generally constant beta value in equation (3)  - line 132.
Response2: Thanks for your nice suggestion. We have added the references for generally constant beta value in equation (3).

We list it as following:

References

  1. Norgett M J, Robinson M T, Torrens I M. A proposed method of calculating displacement dose rates. Nuclear engineering and design, 1975, 33(1), 50-54.

3) Figure 2 why on the Figures reflector starts from 0 up to 170 cm in radius, while in the text it is written that it is 50 cm thick? Please indicate properly reflector and core on all of the Figures. As it is now with marked reflector from 0 up to 170 cm, sudden change of fast neutrons fluxes in at ~150 cm looks strangly.
Response3:Thank you for your reminding. we were really sorry for our careless mistakes. We have corrected the Figure 2 in revised paper, and we have confirmed that the radial of reactor core is 120 cm, and thickness of radial reflector is 50 cm.

“Figure 2 ”have been modified as the foliowing:

(a)

(b)

(c)

“Figure 2. The radial (a) thermal neutron, (b) fast neutron and (c) gamma flux under different barrel materials when the thickness of reflector is 50 cm.”

4) Please write what it estimated uncertainty of the calculations, this can lead to the not smooth behaviour of the calculated values in the figure 5 and 7.

Response4: Thank you for your excellent comments of adding the estimated uncertainty. Based on your suggestions, we have made some modifications in the revised manuscript in paragraph “3.2. The influence of reflector thickness on flux distribution” and “3.3. The influence of boron concentration and distribution on shielding effect”.

Which is list as following:

“In scheme A without barrel, as the reflector becomes thicker, more fast neutrons from active core are moderated into thermal neutrons. On the other hand, the increased thermal neutrons result in more fission reactions in downcomer, which then produce more fast neutrons. Overall, there is a counteraction between the two mechanisms, leading to little change of the fluxes in the outer side of the vessel (Figure 5a). The error of gamma and neutron flux results in Figure 5 is between 3.51% and 4.25%. And the variation tendency of flux (which influenced by reflector thickness) is still obvious, it does not affect the judgment of the results.

“In scheme E, boron is added into 1 cm-thick Hastelloy barrel, and the boron concentration changes from 3% to 30%. As the boron concentration increases, the neutron shielding effect becomes better. However, there is no significant improvement of gamma shielding effect, as shown in Figure 7. Here, the error of neutron and gamma flux results in Figure 7 is betwee 4.08% and 4.45%, To some degree, it could cause the calculation results is not smoothly presented in picture. However, this trend of the fluxes is clear, and it does not affect the judgment of shielding effectiveness.

Round 2

Reviewer 1 Report

The reviewer has examined the revision and reply to reviewer. Most of comments are reflected in the revised manuscript, although model validation n the present study  still remains uncompleted.

Reviewer 4 Report

I am satisfied with the changes of the paper made by authors, and I suggest the paper to be published in the present form.